# Questionnaires and salivary cortisol to measure stress and depression in mid-pregnancy

Richelle Vlenterie[1], Pauline M. Geuijen[1,2], Marleen M. H. J. van Gelder[1,3]*, Nel Roeleveld[1]

**1** Department for Health Evidence, Radboud Institute for Health Sciences, Radboud University Medical Center, Nijmegen, The Netherlands, **2** Department of Psychiatry, Donders Institute for Brain, Cognition and Behavior, Radboud University Medical Center, Nijmegen, The Netherlands, **3** Radboud REshape Innovation Center, Radboud University Medical Center, Nijmegen, The Netherlands

* Marleen.vanGelder@radboudumc.nl

## Abstract

The hypothalamic-pituitary-adrenal axis, with cortisol as its final metabolite, has been proposed as a potential underlying biological mechanism for associations between depression and stress symptoms during pregnancy and adverse perinatal outcomes. In this study, we explored associations between salivary cortisol as a potential biomarker for stress and depressive symptoms and several self-completed psychological measurement scales among pregnant women. In total, 652 pregnant women participating in the PRegnancy and Infant DEvelopment (PRIDE) Study completed the Edinburgh Depression Scale (EDS), Patient Health Questionnaire-2 (PHQ-2), Pregnancy-Related Anxiety Questionnaire-Revised (PRAQ-R), and Tilburg Pregnancy Distress Scale (TPDS) and collected a single awakening salivary cortisol sample around gestational week 17. Odds ratios, Spearman's correlation coefficients ($\rho_s$) and Cohen's Kappa coefficients ($\kappa$) were calculated to examine the associations between the EDS, PHQ-2, PRAQ-R, TPDS, and maternal cortisol levels. The overall correlation coefficient between the score on the EDS and the salivary cortisol level was 0.01 (p = 0.89) with $\kappa$ = -0.01 (95% confidence interval [CI] -0.08–0.06). We did not observe agreement between the PHQ-2 and cortisol levels either ($\kappa$ = 0.06 (95% CI -0.02–0.14)). The results for the PRAQ-R and TPDS were similar with overall correlations with maternal cortisol levels of $\rho_s$ = 0.01 (p = 0.81) and $\rho_s$ = 0.06 (p = 0.35) and agreements of $\kappa$ = 0.02 (95% CI -0.06–0.09) and $\kappa$ = -0.02 (95% CI -0.11–0.07), respectively. Maternal awakening salivary cortisol levels and measures of maternal psychological distress, anxiety, depressive symptoms, and pregnancy-related anxiety, assessed by self-completed questionnaires, did not seem to be related in mid-pregnancy.

## Introduction

Depression and stress disorders during pregnancy have been associated with adverse effects on the health of the mother and the unborn child [1–4]. Therefore, it is important to diagnose

**Data Availability Statement:** Data cannot be shared publicly because it contains sensitive participant information. Furthermore, participants did not give informed consent for data to be made

publicly available in a repository. Data can only be accessed by registered scientists who are authorized to access the data with an individual account and an individual password. Statistical analyses are conducted on a secured server (Digital Research Environment [DRE], www. researchenvironment.org). PRIDE Study data are available upon reasonable request and all requests need approval from the PRIDE Study's Steering Committee. Interested researchers can contact the Chief Secretary of the PRIDE Study Steering Committee (Loes.vanderZanden@radboudumc.nl) to request data access.

**Funding:** This work was supported by the Netherlands Organisation for Health Research and Development (ZonMw; grant number 836011020). The funders had no role in study design, data collection and analysis, decision to publish, or preparation of the manuscript.

**Competing interests:** The authors have declared that no competing interests exist.

these disorders as early in pregnancy as possible. Although the aetiology has not been fully understood yet, disturbances in the hypothalamic-pituitary-adrenal (HPA) axis have been proposed as a potential underlying biological mechanism linking depression and stress symptoms during pregnancy with adverse fetal and child outcomes [5–7]. Cortisol is the final metabolite in the HPA axis and is often related to the state of stress and depression [8]. Especially since cortisol can easily be measured in saliva using a cheap and non-invasive method, it might be a useful biomarker to assess stress and/or depression. Previous studies showed that awakening salivary cortisol levels are reliable biomarkers for measuring an individual's cortisol concentrations compared to levels measured throughout the day [9–11].

For research purposes, stress and depressive symptoms are typically identified by the use of self-administered questionnaires. Several questionnaires exist to assess different aspects of psychological well-being, some of which are also used in clinical settings. Research findings suggest that depression and stress symptoms are associated with elevated diurnal cortisol levels among non-pregnant women [12, 13]. However, this finding cannot be extrapolated straightaway to pregnant women due to pregnancy-related changes in the HPA axis as well as changes in the immune system and hormone levels [14]. This leads to a controversy in the literature about the existence of an association between stress and depressive symptoms assessed through self-administered questionnaires and cortisol levels measured in saliva among pregnant women [15–17]. Depressive symptoms, anxiety disorders, and stress are often associated with similar genes [18, 19], which suggests common underlying neurophysiological mechanisms. However, reported symptoms often differ between the above-mentioned mood disorders [20]. Therefore, we aimed to explore the associations between cortisol level as a potential biomarker and several different self-completed psychosocial measurement scales among pregnant women.

## Materials and methods

### General study design

This study was embedded in a large ongoing prospective cohort study, the PRegnancy and Infant DEvelopment (PRIDE) Study, and was reported according to the STROBE guidelines [21]. The PRIDE Study enrolls Dutch women early in pregnancy. Details of the study design are described elsewhere [22, 23]. In short, pregnant women aged 18 years and older are invited to participate in the PRIDE Study by their midwife or gynecologist just before or during their first prenatal care visit. They are asked to complete Web-based questionnaires at baseline, in gestational weeks 17 (questionnaire 2, Q2) and 34 (questionnaire 3), as well as 2 and 6 months after delivery. Questions concern demographic factors, reproductive history, maternal health, lifestyle factors, and occupational exposures. The prenatal questionnaires also include several measurement scales to assess stress, anxiety, and depressive symptoms. Q2 includes the Edinburgh Depression Scale (EDS), the Patient Health Questionnaire-2 (PHQ-2), and the revised version of the Pregnancy-Related Anxiety Questionnaire (PRAQ-R). These questionnaires were careful selected based on their sensitivity and specificity. The questionnaires are widely used and validated in primary care settings including general and pregnant populations.

For this study, we selected all PRIDE Study participants who completed Q2 between April 2012 and May 2016 (n = 3,027). The PRIDE Study was approved by the Regional Committee on Research involving Human Subjects and all participants provided informed consent.

After completing Q2, participants are asked to donate a single awakening saliva sample. Participants who agree to do so, receive a Salivette® (Sarstedt AG & Co, Nümbrecht, Germany) to collect saliva by regular mail. Samples are taken within 10 minutes after awakening on a working day, before teeth brushing, eating, drinking, or smoking. On the informed

consent form, the women are asked to record the date, time of awaking, and time of saliva collection. All samples are returned to the research site in a special envelope for biological materials by regular mail with a median of 3 days between saliva collection and receipt of the sample.

For the current study, women who participated between December 2014 and May 2016 also received a paper-based version of the Tilburg Pregnancy Distress Scale (TPDS) along with the Salivette®, which we asked to complete on the same day as the saliva sample collection. This questionnaire was included to examine possible effects of the time interval between saliva sampling and self-reporting of depression or stress symptoms in the other three measurement scales administered at least 3 days preceding salivary cortisol collection. The women in this subgroup returned the cortisol sample together with the TPDS questionnaire by regular mail as well.

## Questionnaire characteristics

**Edinburgh Depression Scale.** The EDS is a 10-item self-rating depression scale commonly used to assess depressive symptoms, and is validated for pregnant and postpartum women in the Netherlands [24, 25]. A cut-off value of 10 or greater indicates a positive score for a possible depression on a total scale of 0–30. The EDS consists of two subscales, a depression and a stress subscale.

**Patient Health Questionnaire-2.** The PHQ-2 exists of two questions screening for depressive symptoms. The questions are formulated based on the National Institute for Health and Care Excellence (NICE) guidelines. Patients who screen positive answered at least one question with 'yes' [26–28].

**Pregnancy-Related Anxiety Questionnaire-R.** The PRAQ-R is a shortened version of the PRAQ developed by Van den Bergh [29]. The PRAQ-R is used to assess state and trait anxiety and depression related to pregnancy with a total score range of 10–40 [30]. The 90th percentile cut-off point is used to screen women as positive [31].

**Tilburg Pregnancy Distress Scale.** The TPDS is a questionnaire especially developed to assess pregnancy-related distress among Dutch pregnant women and to pick up negative emotions regarding confinement, delivery, and general health. It includes two sub-scales measuring perceived partner involvement and negative effects. An overall cut-off value of 18 or higher on a total scale of 0–48 indicates a positive score [32].

## Inclusion and exclusion criteria

From the PRIDE Study participants who donated a saliva sample, we excluded all women who did not adhere to the sampling protocol and those who donated an insufficient amount of saliva. In addition, we excluded the samples that were received >14 days after sampling, as well as the samples collected >1 month after completion of Q2. For efficiency reasons, we randomly selected 50% of eligible saliva samples for analysis, with oversampling of samples from women who scored positive on the EDS, PRAQ-R, PHQ-2, and/or TPDS.

## Cortisol assay

Upon receipt at the study site, the saliva samples were immediately stored at -20˚C until analysis. At LDN Labor Diagnostika Nord GmbH & Co. KG, Nordhorn, Germany, the frozen samples were thawed and centrifuged for 5 to 10 minutes at 2000–3000 x g. The Cortisol free in Saliva ELISA Kit (Cortisol Saliva ELISAfree Kit) was used to determine the concentration of cortisol in the samples. Of the standard reagent, the control reagent, and the saliva sample, 50 µL was dispensed in microtiter wells, as well as 50 µL of a cortisol-horseradish peroxidase conjugate for binding to the coated antibody. The wells were incubated for 60 minutes at room temperature. Afterwards, they were rinsed 3 times with 300 µL diluted wash solution

and 200 μL of substrate solution was added. Again, the wells were incubated for 30 minutes at room temperature and 50 μL of stop solution was added to each well. The absorbance of each well was determined with the use of a microtiter plate calibrated reader at 450 ± 10 nm within 15 minutes.

## Statistical analyses

Descriptive statistics were used to describe the general characteristics of the women included in this study. Independent sample t-tests were performed to compare the mean salivary corti-sol levels between the women with positive and negative scores on all questionnaires separately using the standardized cut-off values. In addition, the awakening cortisol levels were dichoto-mized to classify women as having elevated cortisol levels. Currently no clinically relevant threshold value exists, but in a previous study, we showed that the 75th percentile was a reliable cut-off to classify women as having normal or elevated cortisol levels [11]. Logistic regression analysis was used to estimate odd ratios (OR) with 95% confidence intervals (95% CI) for the associations between the dichotomized questionnaire scores and elevated cortisol levels. We calculated the Spearman's rank correlation coefficient ($\rho_s$) to analyze the correlations between the continuous EDS, PRAQ-R, and TPDS scores and the awakening cortisol levels. Cohen's Kappa coefficients ($\kappa$) with 95% confidence intervals were calculated to quantify the degree of agreement between the dichotomized EDS, PRAQ-R, PHQ-2, and TPDS scores and elevated awakening cortisol levels taking agreement occurring by chance into account.

In sub-analyses, we stratified the primary analyses by maternal age ($<$30, 30–34, $\geq$35 years), level of education (low/moderate vs. high), country of birth (The Netherlands vs. other), pre-pregnancy Body Mass Index (BMI; $<$25.0 vs. $\geq$25.0 kg/m$^2$), parity (0 previous births vs. $\geq$1 previous birth), gestational week at completing Q2 (split at the median of 17 weeks), gestational week at saliva sampling (split at the median of 19 weeks), days between completing Q2 and saliva sampling ($\leq$14 days, $>$14 days), awakening time ($\leq$7 a.m. vs. $>$7 a. m.), difference between awakening time and collection time ($\leq$5 min vs. 6–10 min), and infant sex (male vs. female) to examine the correlations and Kappa coefficients among these sub-groups. All statistical analyses were performed using the Statistical Package for Social Sciences (SPSS), version 22.0 for Windows (IBM Corp., Armonk, NY).

## Results

A total of 1,728 out of the 3,019 PRIDE Study participants who completed Q2 in the study period also donated a saliva sample (57.2%), of which 1,304 (75.5%) met the inclusion criteria. Therefore, 652 women were included in this study. The study population included 91 women who scored positive on the EDS, 78 who scored positive on the PRAQ-R, 162 who scored posi-tive on the PHQ-2, and 27 who scored positive on the TPDS. Of note, only 278 women in our study population completed the TPDS.

The average moment of completion of the second PRIDE Study questionnaire, including the EDS, PHQ-2, and PRAQ-R, was gestational week 17 (standard deviation [SD] 1.6), and the average moment of awakening saliva sample collection was gestational week 19 (SD 1.9), mak-ing the average time between completing Q2 and collecting the salivary cortisol sample 15 days (SD 7.2). All women in the subgroup recorded that they completed the TPDS question-naire and collected the saliva sample at the same day.

The mean concentration of cortisol in all samples was 9.98 ng/ml (SD 6.4), whereas the 75th percentile, which was used as cut-off value for elevated cortisol levels, was 12.50 ng/ml. We did not observe differences in the mean cortisol concentrations between women scoring positive on the EDS, PHQ-2, PRAQ-R, and/or TPDS and the women scoring negative on these

**Table 1. Comparison of awakening salivary cortisol levels between women who scored positive and negative on the different measurement scales.** Data from the PRIDE Study, 2012–2016.

| Measurement scale | N | Mean (SD) cortisol level | | Difference (95% CI) | | Elevated cortisol level | | |
| --- | --- | --- | --- | --- | --- | --- | --- | --- |
| | | | | | | N | (%) | OR (95% CI) |
| **EDS** | | | | | | | | |
| Positive | 91 | 9.33 | (4.41) | -0.73 | (-2.15–0.70) | 21 | (23) | 0.9 (0.5–1.6) |
| Negative | 541 | 10.05 | (6.69) | Reference | | 133 | (25) | Reference |
| **PRAQ-R** | | | | | | | | |
| Positive | 78 | 10.13 | (5.00) | 0.14 | (-1.37–1.65) | 21 | (27) | 1.1 (0.7–1.9) |
| Negative | 565 | 9.99 | (6.55) | Reference | | 139 | (25) | Reference |
| **PHQ-2** | | | | | | | | |
| Positive | 162 | 10.39 | (5.45) | 0.52 | (-0.61–1.66) | 48 | (30) | 1.4 (0.9–2.0) |
| Negative | 487 | 9.87 | (6.64) | Reference | | 115 | (24) | Reference |
| **TPDS[a]** | | | | | | | | |
| Positive | 27 | 13.20 | (20.4) | 2.40 | (-5.72–10.45) | 7 | (26) | 0.8 (0.3–2.1) |
| Negative | 251 | 10.80 | (4.9) | Reference | | 74 | (30) | Reference |

[a] The TPDS was not included in PRIDE Study questionnaire Q2, but was completed on the day of saliva sampling by women participating between December 2014 and May 2016.

instruments (Table 1). However, women who scored positive on the PHQ-2 seemed to have an increased risk of an elevated cortisol level (odds ratio 1.4, 95% CI 0.9–2.0).

In Table 2, the correlations between the awakening salivary cortisol levels and the scores on the EDS, PHQ-2, and PRAQ-R are shown. The overall correlation between the EDS scores and the cortisol levels was 0.01 (p = 0.89), while the agreement between both dichotomized measures was κ = -0.01 (95% CI -0.08–0.06). The agreement between the PHQ-2 and the dichotomized cortisol levels was κ = 0.06 (95% CI -0.02–0.14). The PRAQ-R scores and the cortisol levels had an overall correlation of 0.01 (p = 0.81) and an agreement of κ = 0.02 (-0.06–0.09), whereas these values were $\rho_s$ = 0.06 (p = 0.35) and κ = -0.02 (-0.11–0.07) for the TPDS (Table 3).

Stratified analyses performed for all maternal characteristics resulted in similarly low correlation and kappa coefficients (Tables 2 and 3). The strongest correlations were observed among women born outside of The Netherlands (PRAQ-R, $\rho_s$ = 0.41 [p = 0.070]), among overweight/obese women (PRAQ-R, $\rho_s$ = 0.16 [p = 0.038]), and among women pregnant of a male infant (EDS, $\rho_s$ = 0.12 [p = 0.042]). The kappa statistic ranged between -0.17 (95% CI -0.29–0.04) for the PRAQ-R among women aged ≥35 years and 0.41 (95% CI -0.01–0.83) for the PHQ-2 among women born outside of The Netherlands. The analyses on the gestational week of completing the EDS, PHQ-2, PRAQ-R, or TDPS, gestational week of saliva sampling, interval between completion of the questionnaire and saliva sampling (not applicable to the TPDS), awakening time, or difference between awakening time and saliva collection time, did not show any correlations. For these analyses, the kappa statistic ranged between -0.05 (95% CI -0.15–0.05) for the EDS among women who woke after 7am and 0.23 (95% CI -0.15–0.62) for the TPDS among women who completed the PRIDE Study questionnaire (not containing the TPDS) after gestational week 17.

## Discussion

In this study, no correlation was observed between maternal awakening salivary cortisol levels and maternal psychological distress, anxiety or depressive symptoms, and pregnancy-related

**Table 2. Spearman correlation coefficients and Cohen's Kappas of awakening cortisol levels and EDS, PHQ-2, and PRAQ-R scores.** Data from the PRIDE Study, 2012–2016.

| Characteristic | EDS | | | PHQ-2 | | PRAQ-R | | |
|---|---|---|---|---|---|---|---|---|
| | N | Spearman's ρ (p-value) | κ (95% CI) | N | κ (95% CI) | N | Spearman's ρ (p-value) | κ (95% CI) |
| **Overall** | 632 | 0.01 (0.89) | -0.01 (-0.08–0.06) | 649 | 0.06 (-0.02–0.14) | 643 | 0.01 (0.81) | 0.02 (-0.06–0.09) |
| **Maternal age (yrs)** | | | | | | | | |
| <30 | 247 | -0.03 (0.66) | 0.00 (-0.12–0.13) | 254 | 0.11(-0.03–0.24) | 253 | 0.05 (0.46) | 0.05 (-0.08–0.17) |
| 30–34 | 292 | -0.06 (0.34) | -0.01 (-0.11–0.10) | 300 | 0.02 (-0.09–0.14) | 295 | 0.07 (0.27) | 0.07 (-0.04–0.18) |
| ≥35 | 92 | -0.04 (0.70) | -0.05 (-0.22–0.11) | 94 | 0.06 (-0.15–0.26) | 94 | -0.14 (0.18) | -0.17 (-0.29–0.04) |
| **Level of education[a]** | | | | | | | | |
| Low/moderate | 131 | -0.02 (0.80) | -0.02 (-0.18–0.15) | 133 | 0.14 (-0.04–0.32) | 130 | -0.01 (0.92) | 0.04 (-0.13–0.21) |
| High | 497 | 0.02 (0.69) | -0.01 (-0.09–0.07) | 512 | 0.05 (-0.04–0.14) | 509 | 0.04 (0.34) | 0.01 (-0.07–0.09) |
| **Country of birth** | | | | | | | | |
| The Netherlands | 608 | 0.01 (0.83) | -0.01 (-0.09–0.06) | 625 | 0.06 (-0.02–0.14) | 619 | 0.00 (0.92) | 0.02 (-0.06–0.09) |
| Other | 20 | -0.06 (0.79) | 0.17 (-0.28–0.62) | 20 | 0.41 (-0.01–0.83) | 20 | 0.41 (0.07) | 0.23 (-0.28–0.75) |
| **Pre-pregnancy BMI (kg/m²)[b]** | | | | | | | | |
| <25.0 | 466 | -0.01 (0.88) | 0.01 (-0.08–0.09) | 482 | 0.04 (-0.05–0.13) | 479 | -0.05 (0.31) | -0.02 (-0.09–0.06) |
| ≥25.0 | 162 | 0.05 (0.49) | -0.06 (-0.19–0.06) | 163 | 0.13 (-0.03–0.29) | 160 | 0.16 (0.04) | 0.09 (-0.07–0.24) |
| **Parity** | | | | | | | | |
| 0 previous births | 368 | -0.02 (0.71) | -0.06 (-0.14–0.03) | 383 | 0.06 (-0.05–0.16) | 379 | 0.05 (0.34) | 0.03 (-0.07–0.13) |
| ≥1 previous births | 260 | 0.01 (0.82) | 0.03 (-0.09–0.14) | 262 | 0.06 (-0.06–0.18) | 261 | -0.03 (0.68) | 0.01 (-0.07–0.10) |
| **Gestational week of completing Q2** | | | | | | | | |
| ≤17 | 552 | -0.01 (0.91) | -0.04 (-0.11–0.04) | 569 | 0.05 (-0.04–0.13) | 564 | 0.00 (0.99) | 0.02 (-0.06–0.09) |
| >17 | 77 | 0.07 (0.53) | 0.18 (-0.07–0.43) | 77 | 0.22 (-0.02–0.47) | 76 | 0.08 (0.49) | 0.03 (-0.19–0.26) |
| **Gestational week of saliva sampling** | | | | | | | | |
| ≤19 | 467 | 0.03 (0.48) | -0.02 (-0.10–0.07) | 480 | 0.07 (-0.02–0.16) | 476 | 0.00 (0.99) | -0.01 (-0.09–0.07) |
| >19 | 162 | -0.08 (0.30) | 0.01 (-0.13–0.16) | 166 | 0.05 (-0.10–0.21) | 164 | 0.03 (0.70) | 0.09 (-0.06–0.25) |
| **Days between completing Q2 and saliva sampling** | | | | | | | | |
| ≤14 | 337 | 0.02 (0.69) | -0.01 (-0.11–0.09) | 344 | 0.07 (-0.04–0.18) | 342 | 0.02 (0.74) | -0.01 (-0.10–0.09) |
| >14 | 294 | -0.01 (0.84) | -0.02 (-0.12–0.09) | 304 | 0.05 (-0.06–0.17) | 300 | 0.00 (0.96) | 0.04 (-0.06–0.14) |
| **Awakening time** | | | | | | | | |
| ≤7 a.m. | 329 | 0.09 (0.12) | 0.02 (-0.08–0.12) | 337 | 0.08 (-0.03–0.18) | 335 | -0.05 (0.38) | -0.01 (-0.10–0.08) |
| >7 a.m. | 302 | -0.09 (0.14) | -0.05 (-0.15–0.05) | 311 | 0.06 (-0.06–0.17) | 307 | 0.09 (0.13) | 0.06 (-0.06–0.18) |

*(Continued)*

**Table 2.** (Continued)

| Characteristic | EDS | | | PHQ-2 | | PRAQ-R | | |
|---|---|---|---|---|---|---|---|---|
| | N | Spearman's ρ (p-value) | κ (95% CI) | N | κ (95% CI) | N | Spearman's ρ (p-value) | κ (95% CI) |
| **Time between awakening and saliva sampling** | | | | | | | | |
| ≤5 min | 420 | 0.00 (0.95) | -0.01 (-0.10–0.08) | 427 | 0.05 (-0.05–0.15) | 422 | 0.02 (0.63) | -0.02 (-0.10–0.07) |
| 6–10 min | 211 | 0.03 (0.65) | -0.02 (-0.14–0.11) | 221 | 0.09 (-0.05–0.22) | 220 | -0.01 (0.87) | 0.09 (-0.04–0.22) |
| **Infant sex** | | | | | | | | |
| Male | 296 | 0.12 (0.04) | 0.11 (-0.01–0.23) | 304 | 0.16 (0.03–0.28) | 304 | 0.10 (0.08) | 0.05 (-0.06–0.16) |
| Female | 321 | -0.10 (0.09) | -0.13 (-0.21–0.05) | 329 | -0.01 (-0.12–0.10) | 325 | -0.10 (0.09) | -0.01 (-0.11–0.09) |

[b] Low/moderate level of education includes no education, lower general secondary education, intermediate vocational education, higher general secondary education, and pre-university education; High level of education includes higher vocational education and university.

[c] BMI: Body Mass Index

anxiety measured by the EDS, PHQ-2, PRAQ-R, and TPDS, regardless of stratification by maternal characteristics, gestational week at completing the questionnaires, gestational week at saliva sampling, time period between questionnaire completion and saliva sampling, awakening time, or time between awakening and saliva sampling. Therefore, the assumption that psychological characteristics measured by self-completed questionnaires are associated with hyperactivity of the HPA axis, resulting in elevated cortisol levels, is not supported by this study.

Our results are consistent with several other studies that did not find associations between cortisol levels and self-reported measures of psychological functioning among pregnant women either, including studies measuring cortisol levels in multiple saliva samples across a wider gestational window [6, 15, 16, 33, 34] or throughout the day to measure the diurnal rhythm [35], and a study measuring cortisol in serum [36]. Most of these studies had a small sample size, but even in our large cohort, we did not detect these associations. However, other studies did show that cortisol levels were associated with self-reported well-being during pregnancy [17, 37, 38]. Obel et al. [17] reported an association between major life events and evening cortisol levels in late pregnancy, while morning cortisol levels were unaffected. Pluess et al. [37] reported an association between cortisol levels and trait anxiety in 66 women in early, but not in late pregnancy. A third study found an association between an anxiety subscale on cortisol levels at awakening in 170 pregnant women, but none in the overall diurnal cortisol model. They concluded that more research is needed to understand the underlying mechanisms of the association and the search for mechanisms other than the HPA-axis [38]. These studies used different measures of psychological well-being at different time points, finding only a weak association with one of the measures or at one time point, and several nonexistent ones. A recent systematic review by Orta et al. [39] concluded that most studies reported no association between maternal cortisol level and antepartum depression.

The strengths of the current study are the large sample size, the availability of information on many maternal characteristics during pregnancy, and measurements of symptoms of depression, anxiety, and stress by means of several different questionnaires. The EDS, PHQ-2, PRAQ-R, and TPDS are all validated questionnaires measuring different psychological characteristics, such as pregnancy-related anxiety (PRAQ-R), state and trait anxiety (PRAQ-R),

**Table 3. Spearman correlation coefficients and Cohen's Kappas of awakening cortisol levels and Tilburg Pregnancy Distress Scale (TPDS) scores in a subpopulation.**

| Characteristic | TPDS[a] | | |
|---|---|---|---|
| | N | Spearman's ρ (p-value) | κ (95% CI) |
| Overall | 278 | 0.06 (0.35) | -0.02 (-0.11–0.07) |
| **Maternal age (yrs)** | | | |
| <30 | 108 | 0.07 (0.50) | -0.06 (-0.20–0.09) |
| 30–34 | 123 | -0.03 (0.79) | 0.05 (-0.12–0.22) |
| ≥35 | 47 | 0.16 (0.28) | -0.07 (-0.24–0.10) |
| **Level of education[b]** | | | |
| Low/moderate | 49 | -0.02 (0.89) | 0.16 (-0.09–0.42) |
| High | 227 | 0.09 (0.16) | -0.07 (-0.15–0.02) |
| **Country of birth** | | | |
| The Netherlands | 267 | 0.07 (0.29) | -0.01 (-0.11–0.08) |
| Other | 9 | 0.34 (0.38) | -0.13 (-0.30–0.05) |
| **Pre-pregnancy BMI (kg/m$^2$)[c]** | | | |
| <25.0 | 208 | 0.05 (0.50) | -0.05 (-0.14–0.05) |
| ≥25.0 | 69 | 0.08 (0.51) | 0.02 (-0.18–0.21) |
| **Parity** | | | |
| 0 previous births | 153 | 0.08 (0.31) | 0.01 (-0.13–0.15) |
| ≥1 previous births | 124 | 0.01 (0.94) | -0.04 (-0.15–0.07) |
| **Gestational week of completing Q2** | | | |
| ≤17 | 251 | 0.04 (0.50) | -0.04 (-0.14–0.06) |
| >17 | 25 | 0.17 (0.41) | 0.23 (-0.15–0.62) |
| **Gestational week of saliva sampling** | | | |
| ≤19 | 219 | 0.03 (0.63) | -0.04 (-0.15–0.06) |
| >19 | 57 | 0.14 (0.29) | 0.08 (-0.13–0.28) |
| **Awakening time** | | | |
| ≤7 a.m. | 147 | 0.16 (0.05) | 0.00 (-0.11–0.11) |
| >7 a.m. | 131 | -0.05 (0.61) | -0.03 (-0.18–0.12) |
| **Time between awakening and saliva sampling** | | | |
| ≤5 min | 184 | 0.09 (0.22) | -0.02 (-0.14–0.10) |
| 6–10 min | 94 | -0.01 (0.96) | -0.01 (-0.15–0.13) |
| **Infant sex** | | | |
| Male | 128 | 0.13 (0.15) | 0.03 (-0.12–0.17) |
| Female | 144 | 0.01 (0.91) | -0.03 (-0.15–0.09) |

[a] The TPDS was not included in PRIDE Study questionnaire Q2, but was completed on the day of saliva sampling by women participating between December 2014 and May 2016.

[b] Low/moderate level of education includes no education, lower general secondary education, intermediate vocational education, higher general secondary education, and pre-university education; High level of education includes higher vocational education and university.

[c] BMI: Body Mass Index

depressive symptoms (EDS), pregnancy distress (TDPS), and maternal stress (PHQ-2). By adding a subgroup of women completing the TPDS and collecting the salivary cortisol sample at the same day, potential bias due to the time interval between sampling and reporting of symptoms of depression or stress could be ruled out.

A limitation of our study might be the collection of a single salivary cortisol sample, through which we were not able to measure the cortisol awakening response or the circadian

rhythm of cortisol throughout the day. In a previous study we showed that a single awakening salivary sample was reliable to identify pregnant women having elevated cortisol levels [11]. Timing of cortisol assessment is critical, however. Sequential collection in the morning provides information about the cortisol awakening response, whereas evening salivary cortisol provides information on HPA activity. Although challenging regarding feasibility in large study populations, sequential collection throughout the day provides information on the acrophase and amplitude of the circadian rhythm of cortisol. For feasibility reasons, we chose to collect awakening cortisol levels, as salivary cortisol levels in samples collected immediately after awakening have shown a higher day-to-day correlation compared to afternoon samples [10]. Collecting only a single saliva sample in mid-pregnancy, we were not able to examine the associations between psychological functioning and cortisol levels in early or late pregnancy either. Since cortisol levels rise throughout the course of pregnancy [40], we may have missed possible associations in early or late pregnancy. The salivary cortisol levels observed in our study are substantially higher compared to levels in other studies, potentially due to differences in study populations, analytical protocols, and sampling time. Salivary cortisol levels are around twice as high in the morning compared to levels during the day and are two- to three-fold higher in pregnant women compared to non-pregnant women. Averages above 3 ng/mL have been seen in more studies in pregnant women, for example by Jones et al. [10] and Tsubouchi et al. [41]. As we evaluated differences in salivary cortisol levels between groups of participants within our study population, however, any deviations in cortisol levels from other studies did not lead to bias in our results.

In conclusion, we cannot confirm that elevated cortisol levels in pregnant women are associated with positive scores on self-completed questionnaires measuring pregnancy-related anxiety, depressive symptoms, pregnancy distress, and maternal stress. The biological mechanisms underlying the fluctuations in cortisol levels might be different than previously hypothesized, resulting in self-completed questionnaires measuring other aspects. So the role of the complex interplay of factors involved in the hyperactivity of the HPA axis including cortisol remains unclear in depressed or stressed pregnant women. This might suggest that maternal cortisol levels during pregnancy are mainly affected by biological and lifestyle factors, instead of by psychological factors [42]. However, it does not suggest that elevated maternal cortisol levels during pregnancy are harmless, as these have been associated with adverse pregnancy outcomes, such as preterm birth and low birth weight, in previous studies [43, 44]. The etiology and biological mechanisms underlying these associations are not fully explored yet, making this an important topic for future research. Especially during the current COVID-19 pandemic, mental health is particularly relevant given visitation restrictions which complicate participation of family members in the pregnancy trajectory and processes after birth of the child. Longitudinal research is warranted to distinguish patterns of depressive symptoms and stress disorders in pregnant women for targeted interventions.

## Acknowledgments

We would like to acknowledge the mothers and children who continue to take part in this ongoing study.

## Author Contributions

**Conceptualization:** Richelle Vlenterie, Pauline M. Geuijen, Marleen M. H. J. van Gelder, Nel Roeleveld.

**Data curation:** Richelle Vlenterie, Pauline M. Geuijen.

**Formal analysis:** Richelle Vlenterie, Pauline M. Geuijen, Marleen M. H. J. van Gelder.

**Funding acquisition:** Marleen M. H. J. van Gelder, Nel Roeleveld.

**Methodology:** Marleen M. H. J. van Gelder, Nel Roeleveld.

**Supervision:** Nel Roeleveld.

**Writing – original draft:** Richelle Vlenterie.

**Writing – review & editing:** Pauline M. Geuijen, Marleen M. H. J. van Gelder, Nel Roeleveld.

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
