## [Decision Letter · Decision Letter 0]

8 Oct 2020

PONE-D-20-20227

Questionnaires and salivary cortisol to measure stress and depression in mid-pregnancy

PLOS ONE

Dear Dr. van Gelder,

Thank you for submitting your manuscript to PLOS ONE. After careful consideration, we feel that it has merit but does not fully meet PLOS ONE’s publication criteria as it currently stands. Therefore, we invite you to submit a revised version of the manuscript that addresses the points raised during the review process.

We look forward to receiving your revised manuscript.

Kind regards,

Fàtima Crispi

Academic Editor

PLOS ONE

Journal Requirements:

2.We note that you have indicated that data from this study are available upon request. PLOS only allows data to be available upon request if there are legal or ethical restrictions on sharing data publicly. For information on unacceptable data access restrictions, please see http://journals.plos.org/plosone/s/data-availability#loc-unacceptable-data-access-restrictions.

Reviewers' comments:

Reviewer's Responses to Questions

**Comments to the Author**

1. Is the manuscript technically sound, and do the data support the conclusions?

Reviewer #1: Yes

Reviewer #2: Partly

2. Has the statistical analysis been performed appropriately and rigorously? 

Reviewer #1: Yes

Reviewer #2: Yes

3. Have the authors made all data underlying the findings in their manuscript fully available?

Reviewer #1: Yes

Reviewer #2: No

4. Is the manuscript presented in an intelligible fashion and written in standard English?

Reviewer #1: Yes

Reviewer #2: Yes

5. Review Comments to the Author

Reviewer #1: In their secondary cross-sectional analysis of a larger cohort study, Vlenteria et al. produce evaluate the association between maternal awakening salivary cortisol levels and measures of maternal psychological distress, anxiety, depressive symptoms, and pregnancy-related anxiety assessed by self-completed questionnaires completed once during mid-pregnancy. The authors reported that none of the questionnaires used to measure stress and depression in mid-pregnancy were related to maternal prenatal salivary cortisol. This is a clinically important and relevant study because maternal prenatal stress and depression are common problems and is associated with significant morbidity and mortality in both the short- and long-term. Moreover, this study has the potential to represent a significant advancement in the field: while much research has identified prenatal risk factors with likelihood ratios or odds ratios in smaller samples, few studies have attempted to develop reports on easily interpreted associations between maternal prenatal salivary cortisol with stress and depression measures in a sufficiently large sample. These results can have clinical significance in determining which self-reported measures might be given to pregnant woman to facilitate screening for depression and stress disorders during pregnancy, adhering to social distancing, regarding social requirements of the current pandemic. The authors should be applauded for presenting their results in an orderly and simple fashion, without overcomplicating their message (as is often the case in large cohort studies). The statistical methods appear sound and I have no specific concerns although I am not a statistician. I have a lot of enthusiasm about this study and there are few minor issues that I would like to share. I feel these concerns are not sufficient to preclude publication but should be addressed more explicitly by the authors in an effort to make the paper more appealing to a broader audience with novel contributions to this field of research.

Major Comments:

1. I might emphasize the implication of the results of this paper in pregnant populations especially in the period like the one we are living; in the context of COVID 19 pandemic. This information is particularly relevant during the current pandemic given visitation restrictions which complicate participation family members in the pregnancy trajectory and process after the birth of the child. These factors are likely to make mental health consequences more prevalent and severe. One might think that the authors might argue that based on their available data, longitudinal research is warranted to distinguish patterns of depression and stress disorders in pregnant women for targeted interventions to improve depression and stress disorders.

2. The authors do not provide any significant commentary on which self-reported measures might be the best to distribute or implement, and why. Instead, all self-reported questionnaires appear to be treated equally which makes the analysis appear more like data mining. What practical, clinical or statistical considerations might lead to one (or few) of the questionnaires being selected over the others (for further research or clinical application)? How might the self-reported questionnaires be operationalized in clinical practice? What is the underlying clinical plausibility if questionnaires were operationalized? Perhaps this is intended to be the subject of a later paper, but it leaves the clinically oriented reader wanting more.

3. The authors argue that prediction of depression and stress disorders (through self-reported measures) is important to help us target therapies for these disorders. But in fact, most of our current best evidence suggests that conservative preventive strategies including non-pharmacologic interventions and avoidance of certain medications are our most effective therapies. Many of these are probably more easily and effectively implemented as clinic wide practices and are specifically NOT good candidates to be applied selectively. In the future, such therapies may be available, but this should at least be addressed or acknowledged. Otherwise, this looks like a solution in search of problem.

4. Although the sub-analyses that the authors consider and provide are important and relevant, there are some that are missing and require inclusion in this paper (if collected). Please consider providing demographic information and sub-analyses including newborn sex and newborn birth weight, as both these characteristics have shown clinical relation to maternal prenatal depression and stress disorders. Please also confirm if in your analyses post-dates have been excluded.

Minor Comments:

1. In the Introduction on line 54 I would state instead “For research purposes” since “In research settings” elicits thoughts of women attending a particular laboratory or the tests to be administered.

2. Statement on lines 60-62 regarding a controversy in the literature requires a reference.

3. Introduction line 64, provide examples of which mood disorders you refer to and provide please a reference to support this statement.

4. In the methods on line 69 explicitly state this is a secondary analysis of a larger cohort study. I might also consider including the STROBE guidelines as an appendix and stating this the current study under review has been reported to the appropriate STROBE guidelines.

5. Methods line 87, returned to the research site within what time frame? Please clarify.

6. Results line 186, you report an odds ratio with any mention of the methods to conduct this statistical analysis. Please remove this final statement or provide the methods and additional results to support.

7. For most all of the tables, please provide a footnote that defines the level of education being low/moderate of high. Each table should be a standalone contribution to the paper.

8. For most all of the tables, please provide definition for BMI in the footnotes.

9. The footnotes on Table 3 stating that “Q2 did not include the TPDS…” should be included for other relevant tables.

Reviewer #2: The manuscript “Questionnaires and salivary cortisol to measure stress and depression in mid-pregnancy” reports the absence of correlation between awakening salivary cortisol and self-reported questionnaires for depression and anxiety at mid-pregnancy. As such, the study is interesting and the main result (absence of correlation between awakening salivary cortisol and the values of self-reported questionnaires) may add some data to the existing knowledge in the field. However, in my view, the study is presented in a confusing way which may bias the reader. Additionally, several important considerations, e.g. what is the actual meaning of awakening salivary cortisol) should be addressed. In summary, although the topic might be interesting for the field, the presented manuscript contains critical drawbacks which, in my opinion, should be corrected/addressed/modified before being considered for publication in Plos One.

Critical comments:

1. My first critical concern is related with the Introduction. In my opinion, the authors do not present the state of the art of the studied problem. After reading the introduction, the reader might have the impression that there is a solid amount of research supporting the association between depression and high cortisol levels and that this study aims to confirm this evidence. However, in the discussion section, the authors describe several papers (including a review from 2018) reporting no association between maternal cortisol levels and antepartum depression. Therefore, I have doubts about the actual aim of the study. In my opinion, the introduction must present the state of the art of the current knowledge in the field and clearly state to what extent the study will improve this knowledge.

2. Also regarding the introduction, I find that the authors did not select the most accurate literature for their statements. Some examples are:

a) Lines 46-48: “…disturbances in HPA axis have been proposed as a potential underlying mechanism linking depression and stress symptoms during pregnancy with adverse fetal and child outcomes [5,6]”. One would expect that references [5] and [6] will deal with depression, stress and childhood outcomes. However, although both related cortisol with childhood outcomes, none of them are dealing with depression (the main topic of the current paper)

b) Lines 48-49: “cortisol… may be used as a biomarker for the state of stress and depression [7]”. Again, reference [7] does not study cortisol as a marker but the cortisol awakening response (CAR) which requires from different cortisol measurements. In this paper, it states that the association between CAR and awakening hour is different in depression but that does not imply that CAR is a biomarker of depression.

c) Lines 50-52: “Previous studies showed that awakening salivary cortisol levels are reliable biomarkers for measuring an individual’s adrenocortical activity… [7,9]”. Here, the authors mix a paper dealing with CAR, the validation of an at-home collection protocol and the comparison between the levels of one single collection and those from sequential collection. I could not find in these references any study showing that awakening salivary cortisol is a reliable biomarker of adrenocortical activity

d) Lines 56-58: “Research finding suggest that depression and stress symptoms are associated with elevated diurnal cortisol levels among non-pregnant women [10]”. However, reference [10] states already in the abstract “Based on the available studies there is not firm evidence for a difference of salivary cortisol in depressed patients and control persons. Additionally, this reference does not consider stress and differences were reported for both diurnal and evening samples.

I acknowledge that some of these inaccuracies during the introduction do not impact the main message of the study but I strongly suggest the authors to be accurate in their references.

3. I also have a great concern regarding the biological meaning of the awakening salivary cortisol level. The authors use only one measurement of cortisol, and they decided to perform this measurement in the morning in which the concentrations are more time-dependent. It has been already reported (in reference [10]) that “It has been emphasized that infrequent salivary sampling in the morning may fail to detect the pulsative and diurnal pattern of cortisol secretion especially at this time point” and that “Thus, in future studies, it seems mandatory to use frequent measures the first hour after awakening.”. The authors base their selection on their previous study in which they confirmed that one single measurement from one day is equivalent to three single measurements from three different days. In my opinion, the authors should carefully evaluate the meaning of the awakening salivary cortisol level and its limitations.

4. Regarding the limitations of the study, I would include a paragraph detailing the potential limitations of the selection of the awakening salivary cortisol as marker of HPA activity.

Additional comments:

5. Details from the ELISA kit used (company etc) are needed. Other analytical parameters such as limit of quantification and limit of detection of the technique will be also valuable.

6. The reported values for salivary cortisol (average around 10 ng/mL) is higher than the normal values commonly reported (0.2-3 ng/mL). I would expect a discussion about that.

6. PLOS authors have the option to publish the peer review history of their article (what does this mean?). If published, this will include your full peer review and any attached files.

Reviewer #1: No

Reviewer #2: No

---

## [Author Response · Author response to Decision Letter 0]

25 Nov 2020

We highly appreciate the Editors’ and Reviewers’ constructive and useful comments on the manuscript. We adapted several sections of the manuscript in line with these comments. The adapted sections as well as our responses to the comments are described in detail below. The page and line numbers refer to the manuscript version in which the changes are highlighted.

Reviewer's Responses to Questions

Comments to the Author

1. Is the manuscript technically sound, and do the data support the conclusions?

Reviewer #1: Yes

Reviewer #2: Partly

Response: Please see our response to the specific comments raised by Reviewer #2.

2. Has the statistical analysis been performed appropriately and rigorously?

Reviewer #1: Yes

Reviewer #2: Yes

3. Have the authors made all data underlying the findings in their manuscript fully available?

Reviewer #1: Yes

Reviewer #2: No

Response: The Data Availability Statement has been revised:

Data cannot be shared publicly because it contains sensitive participant information. Furthermore, participants did not give informed consent for data to be made publicly available in a repository. Data can only be accessed by registered scientists who are authorized to access the data with an individual account and an individual password. Statistical analyses are conducted on a secured server (Digital Research Environment [DRE], www.researchenvironment.org). PRIDE Study data are available upon reasonable request and all requests need approval from the PRIDE Study's Steering Committee. Interested researchers can contact the study coordinator (Marleen.vanGelder@radboudumc.nl) to request data access.

4. Is the manuscript presented in an intelligible fashion and written in standard English?

Reviewer #1: Yes

Reviewer #2: Yes

5. Review Comments to the Author

Reviewer #1: In their secondary cross-sectional analysis of a larger cohort study, Vlenteria et al. produce evaluate the association between maternal awakening salivary cortisol levels and measures of maternal psychological distress, anxiety, depressive symptoms, and pregnancy-related anxiety assessed by self-completed questionnaires completed once during mid-pregnancy. The authors reported that none of the questionnaires used to measure stress and depression in mid-pregnancy were related to maternal prenatal salivary cortisol. This is a clinically important and relevant study because maternal prenatal stress and depression are common problems and is associated with significant morbidity and mortality in both the short- and long-term. Moreover, this study has the potential to represent a significant advancement in the field: while much research has identified prenatal risk factors with likelihood ratios or odds ratios in smaller samples, few studies have attempted to develop reports on easily interpreted associations between maternal prenatal salivary cortisol with stress and depression measures in a sufficiently large sample. These results can have clinical significance in determining which self-reported measures might be given to pregnant woman to facilitate screening for depression and stress disorders during pregnancy, adhering to social distancing, regarding social requirements of the current pandemic. The authors should be applauded for presenting their results in an orderly and simple fashion, without overcomplicating their message (as is often the case in large cohort studies). The statistical methods appear sound and I have no specific concerns although I am not a statistician. I have a lot of enthusiasm about this study and there are few minor issues that I would like to share. I feel these concerns are not sufficient to preclude publication but should be addressed more explicitly by the authors in an effort to make the paper more appealing to a broader audience with novel contributions to this field of research.

Major Comments:

1. I might emphasize the implication of the results of this paper in pregnant populations especially in the period like the one we are living; in the context of COVID 19 pandemic. This information is particularly relevant during the current pandemic given visitation restrictions which complicate participation family members in the pregnancy trajectory and process after the birth of the child. These factors are likely to make mental health consequences more prevalent and severe. One might think that the authors might argue that based on their available data, longitudinal research is warranted to distinguish patterns of depression and stress disorders in pregnant women for targeted interventions to improve depression and stress disorders.

Response: We fully agree with the reviewer and added a paragraph in the discussion referring to the implications of the study in light of the current pandemic.

Last paragraph of Discussion section (pages 18-19):

Especially during the current COVID-19 pandemic, mental health is particularly relevant given visitation restrictions which complicate participation of family members in the pregnancy trajectory and processes after birth of the child. Longitudinal research is warranted to distinguish patterns of depressive symptoms and stress disorders in pregnant women for targeted interventions.

2. The authors do not provide any significant commentary on which self-reported measures might be the best to distribute or implement, and why. Instead, all self-reported questionnaires appear to be treated equally which makes the analysis appear more like data mining. What practical, clinical or statistical considerations might lead to one (or few) of the questionnaires being selected over the others (for further research or clinical application)? How might the self-reported questionnaires be operationalized in clinical practice? What is the underlying clinical plausibility if questionnaires were operationalized? Perhaps this is intended to be the subject of a later paper, but it leaves the clinically oriented reader wanting more.

Response: After discussion with several epidemiologists, psychologist, and psychiatrists, the EDS, PHQ-2, and PRAQ-R were chosen to be included in the PRIDE Study questionnaires, the large ongoing prospective cohort study in which this study is embedded. The TPDS was administered to a subsample in this study, as this instruments measures other constructs than the EDS, PHQ-2, and PRAQ-R. We added two sentences to the manuscript to elaborate on this. We choose not to include too much detail on the separate questionnaires and their feasibility in clinical practice as this was beyond the scope of this paper.

Lines 82-84:

These questionnaires were careful selected based on their sensitivity and specificity. The questionnaires are widely used and validated in primary care settings including general and pregnant populations.

3. The authors argue that prediction of depression and stress disorders (through self-reported measures) is important to help us target therapies for these disorders. But in fact, most of our current best evidence suggests that conservative preventive strategies including non-pharmacologic interventions and avoidance of certain medications are our most effective therapies. Many of these are probably more easily and effectively implemented as clinic wide practices and are specifically NOT good candidates to be applied selectively. In the future, such therapies may be available, but this should at least be addressed or acknowledged. Otherwise, this looks like a solution in search of problem.

Response: We did not address targeted or any other therapy for depression and stress disorders in this paper. We only argued that early diagnosis (potentially followed by treatment) may be beneficial to prevent adverse effects. We agree with the reviewer that therapy and general preventive strategies should not be applied selectively, so we do not advocate this in the manuscript. 

4. Although the sub-analyses that the authors consider and provide are important and relevant, there are some that are missing and require inclusion in this paper (if collected). Please consider providing demographic information and sub-analyses including newborn sex and newborn birth weight, as both these characteristics have shown clinical relation to maternal prenatal depression and stress disorders. Please also confirm if in your analyses post-dates have been excluded.

Response: We can confirm that we have data available on newborn sex, newborn birth weight, and gestational age. We did not include birth weight and gestational age in our analyses, however, because these our potential outcomes of prenatal depression, stress disorders, and/or increased cortisol levels and are described in another paper (provisionally accepted elsewhere). In this paper, we focus on the associations between maternal self-reported symptoms and maternal cortisol levels during pregnancy, in which gestational age at birth and infant birth weight are very unlikely to play a role. As the activity of the maternal HPA axis varies according to the sex of the foetus, we included this variable in the current study as suggested by the reviewer. 

Minor Comments:

1. In the Introduction on line 54 I would state instead “For research purposes” since “In research settings” elicits thoughts of women attending a particular laboratory or the tests to be administered.

Response: This statement in the introduction was updated according to the recommendation of the reviewer.

Line 54: 

For research purposes, stress and depressive symptoms are typically identified by the use of self-administered questionnaires.

2. Statement on lines 60-62 regarding a controversy in the literature requires a reference.

Response: Three references (already included in the discussion section) were added to support the following statement: This leads to a controversy in the literature about the existence of an association between stress and depressive symptoms assessed through self-administered questionnaires and cortisol levels measured in saliva among pregnant women [15,16,17].

3. Introduction line 64, provide examples of which mood disorders you refer to and provide please a reference to support this statement.

Response: The statement was edited to clarify which mood disorders were referred to. A reference was added to provide support for this statement.

Lines 64-65:

However, reported symptoms often differ between the above-mentioned mood disorders [20].

4. In the methods on line 69 explicitly state this is a secondary analysis of a larger cohort study. I might also consider including the STROBE guidelines as an appendix and stating this the current study under review has been reported to the appropriate STROBE guidelines.

Response: This manuscript does not pertain to a secondary analysis, but to one of the primary studies performed in a large cohort of pregnant women with ongoing recruitment and data collection. Therefore, we did not change the wording on line 69 (now line 71). We did, however, include the STROBE guidelines, as a reference instead of an appendix.

Lines 71-72:

This study was embedded in a large ongoing prospective cohort study, the PRegnancy and Infant DEvelopment (PRIDE) Study, and was reported according to the STROBE guidelines [21].

5. Methods line 87, returned to the research site within what time frame? Please clarify.

Response: The median time frame between saliva collection and receipt of the sample at the research site is 3 days (which was added to the text), with many samples being returned on the day of collection or the day after. As already stated under the inclusion ad exclusion criteria, samples that were received >14 days after sampling were excluded from the analyses.

Lines 94-96:

All samples are returned to the research site in a special envelope for biological materials by regular mail with a median of 3 days between saliva collection and receipt of the sample.

6. Results line 186, you report an odds ratio with any mention of the methods to conduct this statistical analysis. Please remove this final statement or provide the methods and additional results to support.

Response: We thank the reviewer for pointing out this oversight and added logistic regression to the Statistical analyses paragraph. All odds ratios estimated are included in Table 1.

Lines 157-159:

Logistic regression analysis was used to estimate odd ratios (OR) with 95% confidence intervals (95% CI) for the associations between the dichotomized questionnaire scores and elevated cortisol levels.

7. For most all of the tables, please provide a footnote that defines the level of education being low/moderate of high. Each table should be a standalone contribution to the paper.

Response: Footnotes were added to Tables 2 and 3 which define the level of education being low/moderate and high.

Footnotes Tables 2 and 3: 

Low/moderate level of education includes no education, lower general secondary education, intermediate vocational education, higher general secondary education, and pre-university education; High level of education includes higher vocational education and university

8. For most all of the tables, please provide definition for BMI in the footnotes.

Response: The definition for BMI was added to the footnotes of Tables 2 and 3.

Footnotes Tables 2 and 3: 

BMI: Body Mass Index

9. The footnotes on Table 3 stating that “Q2 did not include the TPDS…” should be included for other relevant tables.

Response: The footnote referring to the TDPS not being included in Q2 was also added to Table 1.

Footnotes Tables 1 and 3: 

The TPDS was not included in PRIDE Study questionnaire Q2, but was completed on the day of saliva sampling by women participating between December 2014 and May 2016.

Reviewer #2: The manuscript “Questionnaires and salivary cortisol to measure stress and depression in mid-pregnancy” reports the absence of correlation between awakening salivary cortisol and self-reported questionnaires for depression and anxiety at mid-pregnancy. As such, the study is interesting and the main result (absence of correlation between awakening salivary cortisol and the values of self-reported questionnaires) may add some data to the existing knowledge in the field. However, in my view, the study is presented in a confusing way which may bias the reader. Additionally, several important considerations, e.g. what is the actual meaning of awakening salivary cortisol) should be addressed. In summary, although the topic might be interesting for the field, the presented manuscript contains critical drawbacks which, in my opinion, should be corrected/addressed/modified before being considered for publication in Plos One.

Critical comments:

1. My first critical concern is related with the Introduction. In my opinion, the authors do not present the state of the art of the studied problem. After reading the introduction, the reader might have the impression that there is a solid amount of research supporting the association between depression and high cortisol levels and that this study aims to confirm this evidence. However, in the discussion section, the authors describe several papers (including a review from 2018) reporting no association between maternal cortisol levels and antepartum depression. Therefore, I have doubts about the actual aim of the study. In my opinion, the introduction must present the state of the art of the current knowledge in the field and clearly state to what extent the study will improve this knowledge.

Response: The aim of this study was to examine whether or not associations exist between maternal salivary cortisol levels and self-reported questionnaires scores, as previous studies provided opposite results. We believe that the study improves this field of knowledge by adding more evidence of high quality that this association does not exist and that high levels of cortisol cannot simply be extrapolated to or even replace self-reported outcomes of different questionnaires. To make the introduction more clear, we made the changes as suggested by the reviewer underneath and added several references [15,16,17] that were only addressed in the discussion in the original version of the manuscript. 

2. Also regarding the introduction, I find that the authors did not select the most accurate literature for their statements. Some examples are:

a) Lines 46-48: “…disturbances in HPA axis have been proposed as a potential underlying mechanism linking depression and stress symptoms during pregnancy with adverse fetal and child outcomes [5,6]”. One would expect that references [5] and [6] will deal with depression, stress and childhood outcomes. However, although both related cortisol with childhood outcomes, none of them are dealing with depression (the main topic of the current paper)

Response: A third reference [7] was added which shows a link between hyperactivity of the HPA axis and depression and childhood outcomes.

b) Lines 48-49: “cortisol… may be used as a biomarker for the state of stress and depression [7]”. Again, reference [7] does not study cortisol as a marker but the cortisol awakening response (CAR) which requires from different cortisol measurements. In this paper, it states that the association between CAR and awakening hour is different in depression but that does not imply that CAR is a biomarker of depression.

Response: We added a new reference [8] and updated the text to avoid any strong statements about cortisol being a biomarker for stress and depression.

Lines 47-50:

Cortisol is the final metabolite in the HPA axis and is often related to the state of stress and depression [8]. Especially since cortisol can easily be measured in saliva using a cheap and non-invasive method, it might be a useful biomarker to assess stress and/or depression.

c) Lines 50-52: “Previous studies showed that awakening salivary cortisol levels are reliable biomarkers for measuring an individual’s adrenocortical activity… [7,9]”. Here, the authors mix a paper dealing with CAR, the validation of an at-home collection protocol and the comparison between the levels of one single collection and those from sequential collection. I could not find in these references any study showing that awakening salivary cortisol is a reliable biomarker of adrenocortical activity

Response: The statement in the manuscript was updated to better align with the references mentioned.

Line 50-52:

Previous studies showed that awakening salivary cortisol levels are reliable biomarkers for measuring an individual’s cortisol concentrations compared to levels measured throughout the day.

d) Lines 56-58: “Research finding suggest that depression and stress symptoms are associated with elevated diurnal cortisol levels among non-pregnant women [10]”. However, reference [10] states already in the abstract “Based on the available studies there is not firm evidence for a difference of salivary cortisol in depressed patients and control persons. Additionally, this reference does not consider stress and differences were reported for both diurnal and evening samples.

Response: We thank the reviewer for pointing out this mistake. We replaced ref 10 by two new references ([12,13] that better align with the statement made in the manuscript.

I acknowledge that some of these inaccuracies during the introduction do not impact the main message of the study but I strongly suggest the authors to be accurate in their references.

3. I also have a great concern regarding the biological meaning of the awakening salivary cortisol level. The authors use only one measurement of cortisol, and they decided to perform this measurement in the morning in which the concentrations are more time-dependent. It has been already reported (in reference [10]) that “It has been emphasized that infrequent salivary sampling in the morning may fail to detect the pulsative and diurnal pattern of cortisol secretion especially at this time point” and that “Thus, in future studies, it seems mandatory to use frequent measures the first hour after awakening.”. The authors base their selection on their previous study in which they confirmed that one single measurement from one day is equivalent to three single measurements from three different days. In my opinion, the authors should carefully evaluate the meaning of the awakening salivary cortisol level and its limitations.

Response: In our opinion, we addressed the limitation of using only a single awakening salivary cortisol sample extensively in the discussion. We provided missed possibilities, such as not being able to measure cortisol levels throughout pregnancy and not being able to measure the circadian rhythm of cortisol throughout the day. In this study, we decided to use a single measurement in order to reduce the participation burden and to be able to collect data from a large study population. We hope the reviewer agrees with our extensive paragraph in the discussion.

Existing paragraph in the discussion:

A limitation of our study might be the collection of a single salivary cortisol sample, through which we were not able to measure the cortisol awakening response or the circadian rhythm of cortisol throughout the day. In a previous study, however, we showed that a single awakening salivary sample was reliable to identify pregnant women having elevated cortisol levels. Collecting only a single saliva sample in mid-pregnancy, we were not able to examine the associations between psychological functioning and cortisol levels in early or late pregnancy either. Since cortisol levels rise throughout the course of pregnancy, we may have missed possible associations in early or late pregnancy. 

4. Regarding the limitations of the study, I would include a paragraph detailing the potential limitations of the selection of the awakening salivary cortisol as marker of HPA activity.

Response: A paragraph describing this potential limitation was added to the discussion.

Additional paragraph in the discussion (page 18):

Another potential limitation may be the collection of saliva to measure cortisol levels as marker for HPA activity. Although most studies so far used salivary cortisol as marker of HPA activity, this does not imply that these cortisol levels fully capture HPA activity. Cortisol in blood, hair, or urine may be better markers, but the increased participation burden involved in sampling these materials justifies our choice for salivary cortisol levels as proxy for HPA activity.

Additional comments:

5. Details from the ELISA kit used (company etc) are needed. Other analytical parameters such as limit of quantification and limit of detection of the technique will be also valuable.

Response: The ELISA kits used for the cortisol assays are called Cortisol Saliva ELISAfree kits according to the manual from the LDN Labor Diagnostika Nord GmbH & Co. KG, Nordhorn, Germany, as described in the methods section. We do not have further details on these ELISA kits, but none of the samples had cortisol levels below the detection limit. 

6. The reported values for salivary cortisol (average around 10 ng/mL) is higher than the normal values commonly reported (0.2-3 ng/mL). I would expect a discussion about that.

Response: Thank you for pointing this out. Salivary cortisol levels are two- to three-fold higher in pregnant women compared to non-pregnant women. Morning salivary cortisol levels are also around twice as high as levels during the day. An average above 3 ng/mL has been seen in more studies in pregnant women (e.g., Jones 2006 and Tsubouchi 2011). Differences in study populations and the analytical method may underly our higher observed salivary cortisol levels. However, as we compare differences in levels between groups of participants and not absolute levels, we feel that this did not lead to bias in our study. We added this to the Discussion section:

Lines 295-303:

The salivary cortisol levels observed in our study are substantially higher compared to levels in other studies, potentially due to differences in study populations, analytical protocols, and sampling time. Salivary cortisol levels are around twice as high in the morning compared to levels during the day and are two- to three-fold higher in pregnant women compared to non-pregnant women. Averages above 3 ng/mL have been seen in more studies in pregnant women, for example by Jones et al. [10] and Tsubouchi et al. [41]. As we evaluated differences in salivary cortisol levels between groups of participants within our study population, however, any deviations in cortisol levels from other studies did not lead to bias in our results.

6. PLOS authors have the option to publish the peer review history of their article (what does this mean?). If published, this will include your full peer review and any attached files.

Do you want your identity to be public for this peer review? For information about this choice, including consent withdrawal, please see our Privacy Policy.

Reviewer #1: No

Reviewer #2: No

---

## [Decision Letter · Decision Letter 1]

20 Jan 2021

PONE-D-20-20227R1

Questionnaires and salivary cortisol to measure stress and depression in mid-pregnancy

PLOS ONE

Dear Dr. van Gelder,

Thank you for submitting your manuscript to PLOS ONE. After careful consideration, we feel that it has merit but does not fully meet PLOS ONE’s publication criteria as it currently stands. Therefore, we invite you to submit a revised version of the manuscript that addresses the points raised during the review process.

We look forward to receiving your revised manuscript.

Kind regards,

Fàtima Crispi

Academic Editor

PLOS ONE

Reviewers' comments:

Reviewer's Responses to Questions

**Comments to the Author**

1. If the authors have adequately addressed your comments raised in a previous round of review and you feel that this manuscript is now acceptable for publication, you may indicate that here to bypass the “Comments to the Author” section, enter your conflict of interest statement in the “Confidential to Editor” section, and submit your "Accept" recommendation.

Reviewer #1: All comments have been addressed

Reviewer #2: All comments have been addressed

2. Is the manuscript technically sound, and do the data support the conclusions?

Reviewer #1: Yes

Reviewer #2: Partly

3. Has the statistical analysis been performed appropriately and rigorously? 

Reviewer #1: Yes

Reviewer #2: Yes

4. Have the authors made all data underlying the findings in their manuscript fully available?

Reviewer #1: Yes

Reviewer #2: No

5. Is the manuscript presented in an intelligible fashion and written in standard English?

Reviewer #1: Yes

Reviewer #2: Yes

6. Review Comments to the Author

Reviewer #1: The authors have addressed all of my concerns and I have no remaining comments to the authors. Thank you and congratulations.

Reviewer #2: The authors addressed most of my comments. I really acknowledge the modifications made in the introduction section. However, I still have serious doubts about the meaning of reporting a single cortisol level (irrespective of being collected in saliva or plasma) collected during the morning mainly during pregnancy since it will mix the information of the CAR, the global HPA activity and the hypercortisolemia produced during pregnancy.

In any case, I find that the study is interesting and the authors are honest with their results. For these reasons I advise for publication in PLOS one. However, I still have an important consideration to be modified before publication

Comments:

1. The authors have added in the revised version of the manuscript “Another potential limitation may be the collection of saliva to measure cortisol levels as marker for HPA activity. Although most studies so far used salivary cortisol as marker of HPA activity, this does not imply that these cortisol levels fully capture HPA activity. Cortisol in blood, hair, or urine may be better markers, but the increased participation burden involved in sampling these materials justifies our choice for salivary cortisol levels as proxy for HPA activity.”. I fully disagree with the statement. Salivary cortisol may be the best marker for HPA activity (definitively better than blodd or spot urine) but the timing of the determination is critical. Evening salivary cortisol provides valuable information about HPA activity whereas sequential collections during the morning provide information about the CAR. Sequential collections during the day provide information about the acrophase and the amplitude of the circadian rhythm of cortisol. The limitation of the describes study is not the use of salivary cortisol but the use of the awakening salivary cortisol as a single point. Please, modify this limitation

7. PLOS authors have the option to publish the peer review history of their article (what does this mean?). If published, this will include your full peer review and any attached files.

Reviewer #1: No

Reviewer #2: No

---

## [Author Response · Author response to Decision Letter 1]

2 Feb 2021

We highly appreciate the Editors’ and Reviewers’ time to review our response and amendments made to the manuscript. We adapted one last section of the manuscript in line with the comment of reviewer #2. The adapted section as well as our response to the comment is described below. The page and line numbers refer to the manuscript version in which the changes are highlighted.

Reviewer #1: The authors have addressed all of my concerns and I have no remaining comments to the authors. Thank you and congratulations.

Response: Thank you for this positive remark.

Reviewer #2: The authors addressed most of my comments. I really acknowledge the modifications made in the introduction section. However, I still have serious doubts about the meaning of reporting a single cortisol level (irrespective of being collected in saliva or plasma) collected during the morning mainly during pregnancy since it will mix the information of the CAR, the global HPA activity and the hypercortisolemia produced during pregnancy.

In any case, I find that the study is interesting and the authors are honest with their results. For these reasons I advise for publication in PLOS one. However, I still have an important consideration to be modified before publication:

1. The authors have added in the revised version of the manuscript “Another potential limitation may be the collection of saliva to measure cortisol levels as marker for HPA activity. Although most studies so far used salivary cortisol as marker of HPA activity, this does not imply that these cortisol levels fully capture HPA activity. Cortisol in blood, hair, or urine may be better markers, but the increased participation burden involved in sampling these materials justifies our choice for salivary cortisol levels as proxy for HPA activity.”. I fully disagree with the statement. Salivary cortisol may be the best marker for HPA activity (definitively better than blodd or spot urine) but the timing of the determination is critical. Evening salivary cortisol provides valuable information about HPA activity whereas sequential collections during the morning provide information about the CAR. Sequential collections during the day provide information about the acrophase and the amplitude of the circadian rhythm of cortisol. The limitation of the describes study is not the use of salivary cortisol but the use of the awakening salivary cortisol as a single point. Please, modify this limitation

Response: Thank you for this comment, we indeed inadvertently mixed up the limitations. We fully agree with the reviewer that salivary cortisol may the best biomarker for HPA activity, as opposed to blood or urine as suggested, and that timing of cortisol determination is crucial. The limitation of the use of a single cortisol sample was already extensively addressed in the discussion in lines 289-295, but has now been modified and extended as suggested by the Reviewer. The sentences on other markers have been omitted from the manuscript.

Lines: 292-299

"[…] having elevated cortisol levels [11]. Timing of cortisol assessment is critical, however. Sequential collection in the morning provides information about the cortisol awakening response, whereas evening salivary cortisol provides information on HPA activity. Although challenging regarding feasibility in large study populations, sequential collection throughout the day provides information on the acrophase and amplitude of the circadian rhythm of cortisol. For feasibility reasons, we chose to collect awakening cortisol levels, as salivary cortisol levels in samples collected immediately after awakening have shown a higher day-to-day correlation compared to afternoon samples [10]. Collecting only a single saliva sample in mid-pregnancy, […]"

---

## [Editor Report · Decision Letter 2]

7 Apr 2021

Questionnaires and salivary cortisol to measure stress and depression in mid-pregnancy

PONE-D-20-20227R2

Dear Dr. van Gelder,

We’re pleased to inform you that your manuscript has been judged scientifically suitable for publication and will be formally accepted for publication once it meets all outstanding technical requirements.

Kind regards,

Fàtima Crispi

Academic Editor

PLOS ONE
---

## [Editor Report · Acceptance letter]

14 Apr 2021

PONE-D-20-20227R2 

Questionnaires and salivary cortisol to measure stress and depression in
mid-pregnancy 

Dear Dr. van Gelder:

I'm pleased to inform you that your manuscript has been deemed suitable for publication in PLOS ONE. Congratulations! Your manuscript is now with our production department. 

Kind regards, 

on behalf of

Dr. Fàtima Crispi 

Academic Editor

PLOS ONE